# Structure Learning with Adaptive Random Neighborhood Informed MCMC

**Alberto Caron**[*]
The Alan Turing Institute
London, UK
`acaron@turing.ac.uk`

**Xitong Liang**[*]
Department of Statistical Sciences
University College London
London, UK
`xitong.liang.18@ucl.ac.uk`

**Samuel Livingstone**
Department of Statistical Sciences
University College London
London, UK
`samuel.livingstone@ucl.ac.uk`

**Jim Griffin**
Department of Statistical Sciences
University College London
London, UK
`j.griffin@ucl.ac.uk`

## Abstract

In this paper, we introduce a novel MCMC sampler, PARNI-DAG, for a fully-Bayesian approach to the problem of structure learning under observational data. Under the assumption of causal sufficiency, the algorithm allows for approximate sampling directly from the posterior distribution on Directed Acyclic Graphs (DAGs). PARNI-DAG performs efficient sampling of DAGs via *locally informed*, *adaptive random neighborhood* proposal that results in better mixing properties. In addition, to ensure better scalability with the number of nodes, we couple PARNI-DAG with a pre-tuning procedure of the sampler's parameters that exploits a skeleton graph derived through some constraint-based or scoring-based algorithms. Thanks to these novel features, PARNI-DAG quickly converges to high-probability regions and is less likely to get stuck in local modes in the presence of high correlation between nodes in high-dimensional settings. After introducing the technical novelties in PARNI-DAG, we empirically demonstrate its mixing efficiency and accuracy in learning DAG structures on a variety of experiments.[1]

## 1 Introduction

Structure Learning and Causal Discovery are concerned with reconstructing a graphical model underlying a set of random variables from data, in the form of a Directed Acyclic Graph (DAG), provided that causal identifiability assumptions hold [Pearl, 2009, Drton and Maathuis, 2017, Glymour et al., 2019]. Causal relations can typically be inferred if randomized experiments are carried out [Pearl, 2009]. However, in many relevant applied fields (*e.g.* biology, genomics, ecology, etc.), only observational data are accessible, as performing interventions is usually costly or simply unfeasible. Structure learning from observational data is a challenging problem, with the space of possible DAGs growing super-exponentially in the number of nodes [Robinson, 1977].

**Related work.** Early contributions on structure learning algorithms include *constraint-based methods* [Spirtes et al., 2000, Pearl, 2009], such as PC and FCI, that utilize conditional independence

---

[*]These authors contributed equally to this work.

[1]Code to implement the PARNI-DAG proposal and replicate the experimental sections is available at `https://github.com/XitongLiang/The-PARNI-scheme/tree/main/Structure-Learning`.

37th Conference on Neural Information Processing Systems (NeurIPS 2023).

testing to output a Markov Equivalence Class (MEC). Another stream of literature focuses on *score-based methods* [Geiger and Heckerman, 1994], such as Greedy Equivalent Search (GES) [Chickering, 1996, 2002], which define and consequently maximize a score function associated with different MECs. Alternatively, *functional causal models*, such as LiNGAM [Shimizu et al., 2006] and Additive Noise Models [Hoyer et al., 2008b], directly output DAGs. Recent advancements falling in either of the classes above have considered methods relying on deep learning techniques [Zheng et al., 2018, Yu et al., 2019], but are specifically suited for large sample regimes.

MCMC-based Bayesian approaches, that account for graph uncertainty by learning a posterior distribution over graphs, include Madigan et al. [1995] and Giudici and Castelo [2003], who introduced the MCMC Model Composition (MC$^3$) scheme and the Add-Delete-Reverse (ADR) scheme respectively. Grzegorczyk and Husmeier [2008] improve ADR's mixing by proposing a new arc reversal move. Another stream of contributions on structure learning MCMC have considered working with score functions in smaller spaces rather than the DAG one. Friedman and Koller [2003] first recasted the problem by operating in the space of "orders" instead of DAGs, so that candidate orders, consistent with multiple DAGs, are proposed. Although this generates considerable benefits in terms of scalability, it introduces sampling bias, as some DAGs are over-represented [Ellis and Wong, 2008]. This bias is addressed in Kuipers and Moffa [2017], who proposed working in the space of "ordered partitions" instead, which introduces scalability issues, making partition MCMC applicable to graphs with a small number of nodes. Further works have focused on improving mixing efficiency and scalability of these score-based MCMC [Niinimäki et al., 2016, Viinikka and Koivisto, 2020, Kuipers et al., 2022]. Recent advances have also considered approximate Variational Inference methods [Lorch et al., 2021, Cundy et al., 2021, Geffner et al., 2022]. More closely related to our work, van den Boom et al. [2022] considered the locally-informed proposal on non-decomposable Gaussian Graphical Models (GGM), but with neighborhoods defined by a subset of the space around the current state of the Markov chain, *i.e.* neighborhoods of the MC$^3$ type, with size growing quadratically in the number of nodes. However, their WWA algorithm does not target the posterior on DAGs directly, unlike PARNI-DAG.

Tangential to this work is also the rich literature on discrete spaces MCMC samplers for Bayesian variable selection. Noteworthy contributions include Zanella [2020], that up-weights a random-walk kernel via local neighborhood information and constructs the *locally-informed proposal*, and Griffin et al. [2021], that develops an adaptive MCMC algorithm addressing high-dimensional sparse settings. The locally-informed proposal can be viewed as a discrete analog to the gradient MCMC proposals (*e.g.* the Metropolis-adjusted Langevin algorithm [Roberts and Rosenthal, 1998]), where the gradient of the target distribution does not exist. The adaptive MCMC instead is capable of learning from the shape of the target distribution and exploiting the most important variables. Liang et al. [2022] generalise the locally-informed proposal and the adaptive MCMC to be used within random neighborhood schemes, and introduced the PARNI proposal for Bayesian variable selection.

**Contribution.**    This work presents a novel MCMC sampler for structure learning, **PARNI-DAG**, that builds on top of PARNI [Liang et al., 2022], but is equipped with new features for efficient sampling in the space of DAGs. PARNI-DAG constructs a random neighborhood of possible DAGs via probabilities proportional to a function of the Posterior Edge Probabilities (PEPs), that is, the probabilities of including a direct edge between two nodes. Then, since full enumeration of all the possible DAGs within the neighborhood is virtually impossible in high-dimensions, it proposes a new candidate DAG via a "point-wise update", *i.e.* by considering a sequence of intermediate proposed DAGs belonging to a smaller subset of the neighborhood. Finally, to guarantee better scalability with the number of nodes, we propose a procedure to pre-tune PARNI-DAG's parameters and warm-start the chain by utilising a skeleton graph derived through any constraint or score based algorithm (*e.g.* PC, GES, ...), or the iterative search space expansion proposed in Kuipers et al. [2022].

**Motivations.**    As we will discuss in later sections, PARNI-DAG generates clear advantages over structure MCMC methods such as MC$^3$ and ADR in terms of speed of convergence and mixing, thanks to its new improved proposal, while targeting the same posterior distribution over DAGs. As for the comparison to score-based MCMC methods, it is known that order MCMC [Friedman and Koller, 2003] has better scalability than standard structure MCMC methods, but, operating in the smaller space of order, it introduces sampling bias via a non-uniform prior being assigned to the different DAGs compatible with a single order [Ellis and Wong, 2008]. Compared to order MCMC, PARNI-DAG does not incur in any sampling bias as it targets the posterior on DAGs directly, while

at the same time converging faster than ADR. Kuipers and Moffa [2017] propose a variant called partition MCMC, which operates in the space of ordered partitions instead of orders. Partition MCMC is unbiased in terms of sampling, but it is extremely slow and has high computational complexity (can only be used on very few nodes) [Kuipers et al., 2022]. PARNI-DAG, on the other hand, specifically addresses high-dimensional settings with many nodes.

As a solution to scale order and partition MCMC up to high-dimensional scenarios, the hybrid MCMC scheme of Kuipers et al. [2022] proposes to restrict the initial search space by pruning it via a skeleton graph (*e.g.* PC algorithm derived skeleton) $\mathcal{H}$, whose max parent set size is $m$ per node, so that complexity reduces to $O(n2^m)$. Although this increases MCMC speed dramatically in large graphs, it is likely to introduce bias as some true edges might be deleted in $\mathcal{H}$. To tackle this issue they also propose an iterative procedure (Iterative MCMC) to re-populate $\mathcal{H}$ with additional potential parent nodes, at the expenses of an increased computational time. We couple PARNI-DAG with a similar procedure, but we use a previously derived skeleton $\mathcal{H}$ (e.g. the Iterative MCMC one) not to restrict the space, and possibly cancel out relevant edges, but to warm-start the chain by pre-tuning some of the MCMC, as we will explain in details in later sections.

We show in the experiments section how PARNI-DAG brings about improvements over the classical structure MCMC ADR [Giudici and Castelo, 2003, Grzegorczyk and Husmeier, 2008] and score-based MCMC methods [Friedman and Koller, 2003, Kuipers and Moffa, 2017, Kuipers et al., 2022] in terms of DAG learning accuracy and MCMC mixing, particularly in settings with a high number of nodes, as it is able to reach high probability regions in very few steps, while addressing the presence of highly correlated neighborhoods of edges where classical samplers might get trapped. Code to implement PARNI-DAG and fully reproduce the experiments is provided[2].

## 2 Problem Setup

Consider a graph $\mathcal{G} = (\mathcal{V}, \mathcal{E})$, made of nodes $\mathcal{V}$ and edges $\mathcal{E}$, suitable to represent probabilistic dependencies (edges) between random variables (nodes) through a probabilistic graphical model [Koller and Friedman, 2009]. Given a collection of $n = |\mathcal{V}|$ continuous random variables $(X_1, ..., X_n)$, a Bayesian Network $\mathcal{B} = (\mathcal{G}, \Phi)$ associated with a DAG $\mathcal{G}$, is a probabilistic graphical model utilized to represent a factorization of the joint probability $p(X_1, ..., X_n)$ into the conditional distributions $p(X_1, ..., X_n) = \prod_{i=1}^{n} p_\phi(X_j \mid \mathrm{Pa}(X_j))$, where $\phi \in \Phi$ are the parameters and $\mathrm{Pa}(X_j)$ all the parents of node $X_j$. Suppose that all nodes $(X_1, ..., X_n)$ are re-scaled to have zero mean and unit variance. The goal in structure learning is to learn the BN factorization of conditional probabilities, given a sample of $N$ observations on $\mathcal{D} = \{X_{1,i}, ..., X_{n,i}\}_{i=1}^{N}$. In Bayesian structure learning, we specifically want to learn a posterior distribution on possible DAGs, i.e. $p(\mathcal{G}|\mathcal{D}) \propto p(\mathcal{D}|\mathcal{G})p(\mathcal{G})$. In this work, we define a DAG indicator variable $\gamma \in \Gamma = \{0, 1\}^{n \times n}$, where each $\gamma$ implies a unique graph $\mathcal{G}_\gamma$; $\gamma_{ij} = 1$ indicates the presence of an edge from $X_i$ to $X_j$, while $\gamma_{ij} = 0$ indicates absence of it. The linear functional model associated with DAG $\mathcal{G}_\gamma$ can be specified as

$$X = W_\gamma^\top X + \varepsilon, \quad \text{where } \mathbb{E}(\varepsilon) = \mathbf{0}, \ \mathrm{Var}(\varepsilon) = \mathrm{diag}(\sigma_1^2, ..., \sigma_n^2), \quad (1)$$

and $W_\gamma$ is a matrix of weights. Within a fully Bayesian approach to the inference problem presented in (1), we consider the following prior specification

$$(W_\gamma)_{ij} \mid \sigma_j^2, \gamma_{ij} = 1 \ \sim \ \mathrm{Normal}(0, g\sigma_j^2),$$
$$(W_\gamma)_{ij} \mid \sigma_j^2, \gamma_{ij} = 0 \ \sim \ \delta_0, \qquad p(\sigma_j) \propto \sigma_j^{-2}$$

where $\delta_0$ is the Dirac measure at 0. Thanks to their conjugate form, the coefficients $\{W_{ij}\}_{i,j}$ and the heteroskedastic error variances $\{\sigma_j\}_j$ can be integrated out analytically, leading to a marginal likelihood that depends only on $\gamma$, denoted by $p(\mathcal{D}|\gamma)$. The last building block we need for posterior $\pi(\gamma) \equiv p(\gamma|\mathcal{D})$ inference is acyclicity constraints imposed in the prior distribution for the model indicator $\gamma$, which is specified by

$$p(\gamma) \propto \left(\frac{h}{1-h}\right)^{d_\gamma} \times \mathbb{I}\{\mathcal{G}_\gamma \text{ is a DAG}\} \quad (2)$$

where the hyperparameter $h \in (0, 1)$ represents the prior edge probability and $d_\gamma = |\mathcal{E}_\gamma|$ is the number of edges implied by $\gamma$. We are interested in Bayesian inference on $\gamma$ with posterior distribution $\pi(\gamma) \propto p(\mathcal{D}|\gamma)p(\gamma)$.

---

[2]**Code is provided in the supplementary materials. Github link to be added upon acceptance.**

Notice that the likelihood, and thus posterior distribution, could also be replaced by a scoring function typically used in score-based MCMC methods [Friedman and Koller, 2003, Kuipers and Moffa, 2017, Kuipers et al., 2022], such as the BGe score [Geiger and Heckerman, 2002], or the BDe score, which would make PARNI-DAG suitable also for discrete-valued nodes.

# 3 The novel PARNI-DAG proposal

## 3.1 Point-wise implementation of Adaptive Random neighborhood Informed proposal

Recently, Liang et al. [2022] introduced the Point-wise Adaptive Random Neighborhood Informed proposal (PARNI) in the context of Bayesian variable selection problems. PARNI is characterized by a *Metropolis-Hastings* (MH) proposal [Metropolis et al., 1953, Hastings, 1970] which couples the Adaptively scaled Individual Adaption scheme [Griffin et al., 2021] with the locally-informed proposal [Zanella, 2020]. We begin by firstly describing the generic PARNI proposal for DAG learning problems, then we will introduce the modifications needed to make it efficient and that result in the PARNI-DAG proposal.

The PARNI proposal falls into the class of random neighborhood informed proposals, which are characterized by two steps: i) random sampling of a neighborhood $\mathcal{N}$, and then ii) proposal of a new DAG within this neighborhood $\mathcal{N}$ according to a informed proposal [Zanella, 2020]. The random neighborhoods are drawn based on an auxiliary *neighborhood indicator* variable $k \in \mathcal{K}$, with conditional distribution $p(k|\gamma)$, such that the neighborhood is constructed as a function of $\mathcal{N}(\gamma, k) \subseteq \Gamma$. Let $\mathcal{K} = \Gamma = \{0, 1\}^{n \times n}$, then in PARNI each $k \in \mathcal{K}$ value indicates whether the corresponding position of $\gamma$ is included in the neighborhood. The conditional distribution of $k$ takes a product form $p_\eta(k|\gamma) = \prod_{i,j} p_\eta(k_{ij}|\gamma_{ij})$, characterized by the set of tuning parameters $\eta = \{\eta_{ij}\}_{i,j=1}^n$, where $\eta_{ij} \in (\epsilon, 1 - \epsilon)$ for a small $\epsilon \in (0, 1/2)$, and each $p_\eta(k_{ij}|\gamma_{ij})$ is given by

$$p_\eta(k_{ij} = 1|\gamma_{ij} = 0) = \min\left\{1, \frac{1 - \eta_{ij}}{\eta_{ij}}\right\} , \quad p_\eta(k_{ij} = 1|\gamma_{ij} = 1) = \min\left\{1, \frac{\eta_{ij}}{1 - \eta_{ij}}\right\} . \quad (3)$$

The methods for adapting $\eta$ will be discussed in Section 3.2 specifically. The neighborhood is thus the constructed as

$$\mathcal{N}(\gamma, k) = \left\{\gamma^* \in \Gamma \mid \gamma_{ij}^* = \gamma_{ij} \ \forall (i,j) \ \text{s.t.} \ k_{ij} = 0\right\} . \quad (4)$$

The neighborhood $\mathcal{N}(\gamma, k)$ contains $2^{d_k}$ models where $d_k$ denotes the number of 1s in $k$. The full enumeration over the whole neighborhood is computationally expensive if $d_k$ is big, and infeasible when $d_k$ is beyond 30. Liang et al. [2022] considered a point-wise implementation of the algorithm, which dramatically reduces the computational complexity from $\mathcal{O}(2^{d_k})$ to $\mathcal{O}(2d_k)$. The idea of how the point-wise implementation works is the following. Let variable $K = \{K_r\}_{r=1}^R$ represent the collection of positions where $k_{ij} = 1$. Instead of working with the full neighborhood $\mathcal{N}(\gamma, k)$, we construct a sequence of smaller neighborhoods $\{\mathcal{N}(\gamma(r), K_r)\}_{r=1}^R \subset \mathcal{N}(\gamma, k)$, and a new DAG $\gamma'$ drawn from these neighborhoods $\{\mathcal{N}(\gamma(r), K_r)\}_{r=1}^R$ is sequentially proposed according to the sub-proposals $q_{g,K_r}(\gamma(r-1), \cdot)$ at each time $r$. Given the intermediate DAGs $\gamma = \gamma(0) \to \gamma(1) \to \cdots \to \gamma(R) = \gamma'$, each of their corresponding sub-proposal is defined by

$$q_{g,K_r}(\gamma(r-1), \gamma(r)) = \frac{g\left(\frac{\pi(\gamma(r))p(K_r|\gamma(r))}{\pi(\gamma(r-1))p(K_r|\gamma(r-1))}\right) \mathbb{I}\{\gamma(r) \in \mathcal{N}(\gamma(r-1), K_r)\}}{Z_r} . \quad (5)$$

where $g : [0, \infty) \to [0, \infty)$ is a continuous function and $Z_r$ is the normalising constant. The construction of $K$ and of the sub-neighborhoods $\{\mathcal{N}(\gamma(r), K_r)\}_{r=1}^R$ will be discussed later in Section 3.3.

The choice of function $g(\cdot)$ is crucial to the performance of the PARNI proposal. If $g(x) = 1$, the sub-proposal $q_{g,K_r}$ in (5) reverts back to a random walk proposal which proposes a new DAG from $N(\gamma(r-1), K_r)$ with uniform probability. It would then be hard for it to explore important neighborhoods of edges in high-dimensions, as it is likely to get stuck in neighborhoods with highly correlated nodes. In the locally-informed proposal of Zanella [2020], function $g(\cdot)$ is chosen to be a non-decreasing and continuous balancing function $g : [0, \infty) \to [0, \infty)$, which satisfies $g(x) = xg(1/x)$. Zanella [2020] shows that the locally-informed proposal with balancing function $g(\cdot)$ is asymptotically optimal compared to other choices of $g(\cdot)$ in terms of Peskun ordering. In this

work, we consider the Hastings' choice (*i.e.* $g(x) = \min\{1, x\}$, satisfying $g(x) = xg(1/x)$) as it has been shown to be the most stable choice for most problems (see the discussion in Supplement B.1.3 of Zanella [2020]).

The informed proposal in this context not only helps speed up convergence, but it also avoids proposing invalid DAGs. This is because the acyclicity checks for candidate models $\gamma' \in \mathcal{N}(\gamma(r - 1), K_r)$ can be run before computing their posterior model probabilities, and thus one can assign zero mass to their prior (resulting in zero mass posterior) if $\gamma'$ is not a valid DAG. The resulting proposal kernel is then given by

$$q_{g,K}(\gamma, \gamma') = \prod_{r=1}^{R} q_{g,K_r}(\gamma(r - 1), \gamma(r)). \tag{6}$$

In order to construct a $\pi$-reversible chain and calculate the MH acceptance probability, we define the new collection of variables $K' = \{K'_r\}_{r=1}^{R}$ for the reverse moves, where $K'$ contains the same element as $K$ but with reverse order (*i.e.* $K'_r = K_{R-r+1}$). The MH acceptance probability is then defined as

$$\alpha(\gamma, \gamma') = \min\left\{1, \frac{\pi(\gamma')q_{g,K'}(\gamma', \gamma)}{\pi(\gamma)q_{g,K}(\gamma, \gamma')}\right\}. \tag{7}$$

**Proposition 3** of Liang et al. [2022] shows that using the above $K'$ can simplify the calculation of the MH acceptance probability in (7).

In the following sections, we will introduce the necessary modifications and developments to efficiently sample DAGs, the combination of which defines the novel PARNI-DAG proposal. The full details and algorithmic pseudo code of PARNI-DAG are also provided in the supplementary material (Appendix D).

## 3.2 Warm-start of $\eta$ combined with hybrid scheme

Griffin et al. [2021] and Liang et al. [2022] both suggested using the Posterior Inclusion Probabilities in a Bayesian variable selection context to define the $\eta_{ij}$ hyperparameters described above. Perfectly analogous, we will make use of the Posterior Edge Probabilities (PEPs) instead, *i.e.* $\pi(\gamma_{ij} = 1)$. The distribution in (3) with the choice of parameters described, encourages to 'flip' edge values based on variables' marginal importance. For example, suppose the posterior for $\gamma_{ij}$ is $\pi(\gamma_{ij} = 1) = 0.9$, and the chain is currently at $\gamma_{ij} = 0$. The edge should be then assigned higher probability to be included in the DAG (*i.e.* $\gamma_{ij}$ should be switched from 0 to 1). Thus, we propose $k_{ij} = 1$ with probability $\min\{1, 0.9/0.1\} = 1$, so that the new neighborhood features this possible move and $\gamma_{ij}$ can be effectively switched from 0 to 1 in the next moves. Another advantage of using the conditional distribution featuring the PEPs in (3) is that when all variables are independent this proposal will have acceptance probability 1 and achieve optimal asymptotic variance as showed in Griffin et al. [2021].

The PEPs, $\pi(\gamma_{ij} = 1)$, however cannot be directly calculated for most problems. Griffin et al. [2021] considered the Rao-Blackwellised estimates $\hat{\pi}(\gamma_{ij} = 1|\gamma_{-ij})$, where $\gamma_{-ij}$ denote $\gamma$ excluding the edge indicator $\gamma_{ij}$, and $\eta$ is updated on the fly given the current output of the chain. The Rao-Blackwellisation is however infeasible in the context of DAG sampling because it entails computing $n^2$ model probabilities at each iteration of the MCMC. This would result in an impractical computational scheme when working with many nodes. Alternatively, one could take the ergodic sum over the output $\{\gamma^{(l)}\}_{l=1}^{t}$ as

$$\hat{\pi}_{ij}^{(t)} = \frac{1}{t} \sum_{l=1}^{t} \mathbb{I}\left\{\gamma_{ij}^{(l)} = 1\right\}, \tag{8}$$

but the ergodic sum would be too slow to converge. This updating scheme would also lead to other issues such as poor exploration and mis-specification of the important edges. It is worth mentioning that the $\eta$'s do not have to be the exact PEPs, as we can still draw valid samples from the target posterior distribution $\pi$ as long as the MH acceptance probability preserves $\pi$-reversibility. However, choosing $\eta$'s that are close to the true PEPs will cause each component to change and to target the right proportion of their marginal probabilities (higher probability of including important edges).

Our solution then is to approximate the PEPs $\tilde{\pi}_{ij}$ before actually running the PARNI-DAG's MCMC, and use this approximation to warm-start the chain. While running the algorithm at time $t$, we update

the tuning parameters $\hat{\eta}_{ij}^{(t)}$ adaptively based on

$$\hat{\eta}_{ij}^{(t)} = \phi_t \tilde{\pi}_{ij} + (1 - \phi_t)\hat{\pi}_{ij}^{(t)} , \qquad (9)$$

where $\{\phi_l\}_{l=1}^t$ is a decreasing sequence of weights which control the trade-off between the provided warm-start PEPs $\tilde{\pi}_{ij}$ and the ergodic sum of the output $\hat{\pi}_{ij}^t$. The choice of $\{\phi_l\}_{l=1}^t$ is not unique and we give a full specification of our choice in the supplementary material (Appendix B). In what follows, we will briefly describe the methods used to calculate the warm-start approximation $\tilde{\pi}_{ij}$.

We argue that the iterative procedure used to restrict the initial search space in Kuipers et al. [2022] can be particularly useful in our case to efficiently approximate the PEPs before running the chain. Following the same notation as in Kuipers et al. [2022], let $\mathcal{H}$ be the adjacency matrix underlying a skeleton graph obtained from any DAG learning algorithm (*e.g.* PC, GES, etc.). For node $X_j$, the adjacency matrix $H$ defines the *permissible parent set* $h^j = \{X_i : H_{ij} = 1\}$. All possible parent combinations of node $X_j$ are included in the *collection of permissible parent set* which is defined by

$$\mathbf{h}^j = \{m| \ m \subseteq h^j\}. \qquad (10)$$

The collection of permissible parent set $\mathbf{h}^j$ can be then extended to a bigger collection $\mathbf{h}_+^j$ as described in Section 4.3 of Kuipers et al. [2022]. The extended collection of permissible parent set $\mathbf{h}_+^j$ essentially includes one additional node from outside the permissible parent set $h^j$ as an additional parent, otherwise the candidate parents of $X_j$ are restricted to the set $h^j$. After deriving $\mathbf{h}_+^j$ via this procedure as in Kuipers et al. [2022], we then approximate $\tilde{\pi}_{ij}$ through the following procedure:

1. We remove the acyclicity constraint from the prior (2) and compute the unconstrained posterior distribution $\pi^u(\gamma)$, approximating the marginal probability $\pi^u(\gamma_{ij})$ under $\mathbf{h}_+^j$.

2. We approximate the joint probability of a pair edges by the product

$$\tilde{\pi}(\gamma_{ij}, \gamma_{ji}) \approx \pi^u(\gamma_{ij}) \times \pi^u(\gamma_{ji}). \qquad (11)$$

3. We impose the simplest acyclicity constraint which avoids creating two edges with opposite directions, *i.e.* let $\tilde{\pi}(\gamma_{ij} = 1, \gamma_{ji} = 1) = 0$. Approximate

$$\tilde{\pi}_{ij} = \frac{\tilde{\pi}(\gamma_{ij} = 1, \gamma_{ji} = 0)}{\tilde{\pi}(\gamma_{ij} = 1, \gamma_{ji} = 0) + \tilde{\pi}(\gamma_{ij} = 0, \gamma_{ji} = 1) + \tilde{\pi}(\gamma_{ij} = 0, \gamma_{ji} = 0)}. \qquad (12)$$

The full details for the calculations of $\tilde{\pi}_{ij}$ are given in the supplementary material (Appendix C).

### 3.3 Introducing the "reversal" neighborhood

In the original PARNI proposal for Bayesian variable selection, each $K_r$ in the intermediate proposals corresponds to exactly one position $(i, j)$, such that $k_{ij} = 1$ and that each intermediate neighborhood $\mathcal{N}(\gamma(r-1), K_r)$ only contains two DAGs: $\gamma(r-1)$ itself and the DAG obtained by flipping $K_r$ in $\gamma(r-1)$. This sequential neighborhood construction is clearly not appropriate for the problem of DAG sampling and might result in very slow mixing. In addition, it is natural to assume that a pair of edges $i \longleftrightarrow j$ are highly correlated and thus equally likely to be included in a DAG. It is then necessary to specify a reversal move that achieves good mixing between pairs of edges. The neighborhood construction in the original PARNI proposal does not support a reversal move in one single sub-proposal. To elaborate on this, suppose that currently $\gamma_{ij} = 1$ and $\gamma_{ji} = 0$, to propose a reversal move according to the original PARNI proposal, we should first propose to remove $\gamma_{ij}$ from the DAG, then include $\gamma_{ji}$ in the following sub-proposals. These flips might be running into issues if the posterior model probability for excluding both edges, $\gamma_{ij} = \gamma_{ji} = 0$, is low, so that these moves are rarely proposed and the chain gets stuck at $\gamma_{ij} = 1$.

Therefore, we construct a bigger neighborhood containing the edge reversal move of a DAG. This modification allows for an edge reversal move within a single sub-proposal $q_{g,K_r}$ without going through the low posterior probability models and avoids the chain getting stuck. To clarify this, suppose $k$ is sampled at each iteration of the chain. For all those pairs which satisfy $k_{ij} = 1$ and $k_{ji} = 1$ in $k$, we construct the reversal neighborhoods for edges $\gamma_{ij}$ and $\gamma_{ji}$. For other positions such that $k_{ij} = 1$, but $k_{ji} = 0$, we construct the original neighborhood which only contains the current model and model with $\gamma_{ij}$ flipped. This reversal neighborhood specification is included into the main PARNI-DAG proposal given in the supplementary material (Appendix D).

### 3.4 New adaptive scheme for better computational efficiency

Although the point-wise implementation reduces the per-iteration costs from exponential to linear in the neighborhood size (equivalent to the number of sub-neighborhoods in PARNI), the PARNI proposal is still computationally expensive when the sampled DAGs are not very sparse. For example, considering the `gsim100` simulated data of Suter et al. [2023], featuring 161 true edges, we would be often required to compute 320 posterior DAG probabilities on average in a single MCMC iteration, and the computation thus becomes 320 times more expensive than the standard ADR proposal in this case. To reduce the computational cost of the PARNI-DAG proposal even more, we make use of a 'neighborhood thinning' parameter $\omega$ to control the number of neighborhoods to be evaluated.

In the original PARNI proposal, the thinning parameter $\omega$ represents the random walk jumping probability, so it does not take the posterior inclusion probabilities into account and does not affect the number of neighborhoods evaluated. For the new PARNI-DAG proposal instead, with probability $1 - \omega$, we will skip evaluation of some of the neighborhoods. Within each iteration, for each point $r$ in the sequence, given the current sub-proposal $\gamma(r - 1)$ and sub-neighborhood indicator $K_r$, PARNI-DAG:

1. with probability $\omega$, proposes $\gamma(r) \sim q_{K_r}(\gamma(r-1), \cdot)$ as in (5).
2. with probability $1 - \omega$, does not evaluate the posterior probabilities of other DAGs in $\mathcal{N}(\gamma(r-1), K_r)$, and directly propose $\gamma(r) = \gamma(r-1)$.

Varying $\omega$ can vary the number of posterior DAG probability evaluations. The larger the value of $\omega$, the more the DAG probabilities evaluated. If $\omega = 1$, all neighborhoods are required to be evaluated to propose a new DAG.

The parameter $\omega$ is updated adaptively. Here, we consider the Robins-Monro adaptation scheme to update $\omega$ on the fly. At time $t$, the new $\omega^{(t+1)}$ parameter is updated via

$$\text{logit}_\epsilon \omega^{(t+1)} = \text{logit}_\epsilon \omega^{(t)} - \psi_t(\mathcal{N}_t - \tilde{\mathcal{N}}) \tag{13}$$

where $\mathcal{N}_t$ is number of sub-neighborhoods evaluated at time $t$ and $\tilde{\mathcal{N}}$ is a pre-specified expected number of evaluations. We set $\psi_t = t^{-0.7}$, meaning that the adaptation of $\omega$ diminishes at the rate of $\mathcal{O}(t^{-0.7})$. From the empirical results, we find that $\tilde{\mathcal{N}} = 10$ evaluated neighborhoods on average seems to achieve optimal balance between mixing performance and computational cost. Therefore, we choose to set $\tilde{\mathcal{N}} = 10$ for most of the numerical studies in the next section and in the supplementary material.

## 4 Experiments

### 4.1 Convergence and mixing efficiency

**Protein data.** We first consider the real-world protein-signalling dataset [Sachs et al., 2005], found also in Cundy et al. [2021], to test PARNI-DAG's mixing. The dataset consists of $n = 11$ nodes and $N = 853$ observations and the ground-truth DAG (17 edges) is provided by expert knowledge. We compare the performance of PARNI-DAG to ADR, Order MCMC and Partition MCMC. We provide the full details on the implementation of the compared models in the supplementary materials (Appendix E). For a fair comparison of the models, we re-define the score functions for Order and Partition MCMC such that all schemes target the same posterior distribution as specified in Section 2. Notice that for this dataset, it is feasible for all the four MCMC schemes to use the full skeleton, without the need to restrict the initial space [Kuipers et al., 2022]. For ADR and PARNI-DAG, we use prior parameters $g = 10$ and $h = 1/11$.

We compare trace plots of log posterior DAG probabilities for ADR, PARNI-DAG and Partition MCMC in Figure 1. Since we could not extract the full score trace of Order MCMC using the `BiDAG` package [Suter et al., 2023], Order MCMC is not included in Figure 1. We ran ADR, PARNI-DAG and Partition MCMC for 480,000, 60,000 and 20,000 iterations respectively which take about the same CPU time[3], and thin the output from ADR and PARNI-DAG so that 20,000 measures are kept. From Figure 1, all these three algorithms mix equally well, the only difference is that the Partition

---

[3]Using Intel i7 2.80 GHz processor

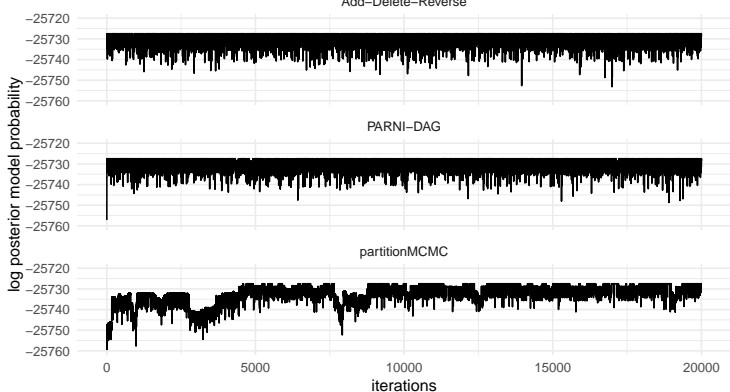

Figure 1: Protein dataset: trace plots of log posterior model probabilities. All algorithms were ran for the same CPU-time and thinned to 20,000 measures.

| algorithm | MSE |
|---|---|
| **ADR** | 8.84 |
| **PARNI-DAG** | 19.44 |
| **Order MCMC** | 145.17 |
| **Partition MCMC** | 121.83 |

Table 1: Protein dataset: time-normalised median MSE $(\times 10^{-8})$ on estimating posterior edge probabilities. (*Lower is better.*)

MCMC algorithm explores more DAGs with lower probabilities. We then compare the median mean squared errors (MSE) on estimating PEPs compared to a ground-truth estimate. The MSE assesses both the convergence of the chain and the bias in the Monte Carlo estimates. The ground-truth PEPs were obtained by running Partition MCMC for approximately 5 hours. We ran all algorithms for 20 replications. Each individual chain was run for 3 minutes. The MSE estimates are presented in Table 1. The structure MCMC schemes (ADR and PARNI-DAG) generally outperform Order MCMC and Partition MCMC and return estimates with lower MSE. Notice that ADR in this specific case also outperforms PARNI-DAG, since the dataset is very low-dimensional (11 nodes), with PARNI-DAG not trailing very far behind. We will demonstrate in the next experiments how PARNI-DAG instead performs better in higher-dimensional settings.

**gsim100 data.** We also study the performance of ADR, PARNI-DAG, Order MCMC and Partition MCMC on a more complex graph, the `gsim100` simulated dataset, found in the `BiDAG` package [Suter et al., 2023]. This features a randomly generated DAG with $n = 100$ nodes and $N = 100$ observations, with 161 true edges. For ADR and PARNI-DAG we set prior parameters to $g = 10$ and $h = 1/100$. Since $n > 20$, it is not feasible to use the full skeleton as in the protein dataset. So we consider two skeletons to restrict the initial space: PC derived skeleton $\mathcal{H}_{PC}$ and the skeleton derived with the iterative procedure of Kuipers et al. [2022] $\mathcal{H}_{iter}$, featuring more edges than $\mathcal{H}_{PC}$.

We examine the trace plot of log posterior DAG probabilities from ADR, PARNI-DAG and Partition MCMC using both skeletons $\mathcal{H}_{PC}$ and $\mathcal{H}_{iter}$ respectively (Figure 2). We ran ADR, PARNI-DAG and Partition MCMC for 80,000, 20,000 and 20,000 iterations respectively. Notice that, under both $\mathcal{H}_{PC}$ and $\mathcal{H}_{iter}$, PARNI-DAG converges very quickly to the high-probabilistic region, whereas ADR takes longer. Partition MCMC under $\mathcal{H}_{PC}$ cannot reach the same high-probability region as it targets a posterior distribution restricted to $\mathcal{H}_{PC}$, which is biased if $\mathcal{H}_{PC}$ excludes true positive edges. Under the more populated $\mathcal{H}_{iter}$ instead it converges, but still slower than PARNI-DAG. When comparing median MSEs on PEPs on this dataset, we considered 20 replications with each individual chain being run for 15 minutes. As for Order and Partition MCMC, $\mathcal{H}_{PC}$ seems to lead to slightly better performance than $\mathcal{H}_{iter}$. This might be due to the fact that in this high-dimensional example $\mathcal{H}_{iter}$ includes considerably more DAGs than $\mathcal{H}_{PC}$, so Order MCMC, and even more so Partition MCMC, take longer to explore the space. As for ADR and PARNI-DAG, the choice of $\mathcal{H}$ does not have a big impact, as in their case this is used only to warm-start the chain and not to restrict the space. PARNI-DAG here outperforms ADR and it is competitive to Order MCMC as well, without introducing biases stemming from search space restriction and sampling.

## 4.2 DAG learning accuracy

Lastly, we compare performance of PARNI-DAG with other main methods in recovering the underlying true DAG on four different real-world graph structures. These graph structures are taken from the

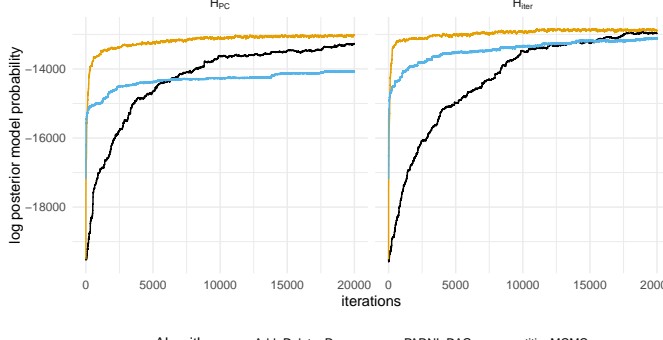

| skeleton | algorithm | MSE |
|---|---|---|
| $\mathcal{H}_{\mathrm{PC}}$ | **ADR** | 2.91 |
| | **PARNI-DAG** | 1.83 |
| | **Order MCMC** | 1.79 |
| | **Partition MCMC** | 12.75 |
| $\mathcal{H}_{\mathrm{iter}}$ | **ADR** | 2.37 |
| | **PARNI-DAG** | 1,72 |
| | **Order MCMC** | 1.84 |
| | **Partition MCMC** | 31.16 |

Figure 2: `gsim100` dataset: trace plots of log posterior model probabilities. All algorithms were ran for the same CPU-time and thinned to 20,000 measures.

Table 2: table `gsim100` dataset: time-normalised median MSE ($\times 10^{-7}$) on estimating posterior edge probabilities. (*Lower is better.*)

R package `bnlearn`[4] and include: i) *ecoli70* dataset, with $n = 46$ nodes and 70 edges; ii) *magic-niab* dataset, with $n = 44$ nodes and 66 edges; iii) *magic-irri* dataset, with $n = 64$ nodes and 102 edges; iv) *arth150* dataset, with $n = 107$ nodes and 150 edges. Given the graph structure and its associated parameters, we simulate $N = 100$ *i.i.d.* observations for each of the four cases. The models we compare include: i) **PC** algorithm; ii) **GES**; iii) **LiNGAM**; iv) **DiBS**, a differentiable variational method introduced by Lorch et al. [2021]; v) **DiBS**+, a version of DiBS with the weighted particle mixture; vi) **DECI**, another non-linear additive noise model that uses variational inference introduced by Geffner et al. [2022]; vii) **Order MCMC**, on a restricted space defined by the skeleton of the PC algorithm; viii) **Iterative MCMC**, the version of Order MCMC run on a search space defined by starting with the PC skeleton, and iteratively expanding the possible parent sets until the score cannot be improved [Kuipers et al., 2022]; ix) **Partition MCMC**, on the same space defined by the iterative procedure in Iterative MCMC; x) **ADR** scheme and xi) **PARNI-DAG**, both with pre-tuning of the $\eta$'s using the Iterative-derived skeleton. All the score-based methods (GES, Order, Partition, Iterative) are assigned an edge-penalty parameter equal to $2log(n)$, as suggested in [Kuipers et al., 2022], to adjust to potentially sparse graphs. Performance is measured via the Structural Hamming Distance between the true DAG and the estimated DAG, averaged over 20 replications of the experiment, for each dataset. Results are collected in Table 3. Notice how PARNI-DAG is either the best or the second best method across all four experiments. In particular, it statistically matches the performance of the resulting best MCMC scheme in the first three experiments featuring a lower number of nodes, while it significantly outperforms all the methods in the case with high number of nodes (*arth150*), as expected. All variational methods (DiBS, DiBS+ and DECI) struggle to detect edges in these low-sample setups as they require a stronger signal (or a larger sample, as featured in the experiments in Lorch et al. [2021], Geffner et al. [2022]) to perform well. Additionally, they are significantly slow to converge and run into numerical instability issues in the last large-$n$ dataset (*arth150*).

## 5 Limitations

Although the PARNI-DAG proposal has demonstrated significant improvements over the state-of-the-art algorithms considered in the previous section, it still comes with a few limitations that can be addressed. The first limitation lies in the linearity assumption. In fact, in order to derive the closed-form marginal likelihood featured in the locally informed proposal, PARNI-DAG is currently constrained to the linear model assumption described in (1). This can potentially result in model misspecification in presence of non-linear relationships, which is a threat in cases where this hampers edges identifiability. The recent work of Liang et al. [2023] has shed light on how to efficiently extend the PARNI proposal to more general non-linear models, and can possibly be adjusted to Bayesian structure learning settings that employ Additive Noise Models (ANMs) [Hoyer et al., 2008a], where a closed-form marginal likelihood does not exist. The second limitation of the PARNI-DAG proposal is the computational cost associated with the likelihood evaluation, which scales at least linearly in the

---

[4]More information on: http://bnlearn.com/.

|                | ecoli70          | magic-niab       | magic-irri        | arth150           |
| :------------: | :--------------: | :--------------: | :---------------: | :---------------: |
| **PC**         | $62.65 \pm 1.64$ | $66.95 \pm 0.82$ | $103.75 \pm 0.91$ | $126.70 \pm 2.10$ |
| **GES**        | $47.95 \pm 1.72$ | $65.50 \pm 0.42$ | $97.60 \pm 0.99$  | $129.70 \pm 3.47$ |
| **LiNGAM**     | $107.4 \pm 3.96$ | $71.60 \pm 1.34$ | $111.25 \pm 2.34$ | -                 |
| **DiBS**       | $71.05 \pm 0.17$ | $65.80 \pm 0.24$ | $102.35 \pm 0.17$ | -                 |
| **DiBS+**      | $71.25 \pm 0.28$ | $64.70 \pm 0.36$ | $100.45 \pm 0.40$ | -                 |
| **DECI**       | $70.95 \pm 0.02$ | $66.95 \pm 0.02$ | $102.00 \pm 0.00$ | -                 |
| **Order MCMC**     | $42.85 \pm 1.30$       | $\mathbf{62.05 \pm 1.52}$ | $\mathbf{92.90 \pm 1.46}$ | $110.00 \pm 1.96$       |
| **Iterative MCMC** | $\mathbf{39.55 \pm 2.15}$ | $65.00 \pm 2.43$       | $96.15 \pm 2.11$       | $119.45 \pm 3.43$       |
| **Partition MCMC** | $48.85 \pm 2.76$       | $66.20 \pm 1.62$       | $103.35 \pm 1.81$      | $140.75 \pm 4.04$       |
| **ADR**            | $\mathbf{39.40 \pm 1.27}$ | $64.25 \pm 0.64$       | $95.20 \pm 1.16$       | $143.15 \pm 1.62$       |
| **PARNI-DAG**      | $\mathbf{37.55 \pm 1.59}$ | $\mathbf{64.14 \pm 0.55}$ | $\mathbf{91.85 \pm 0.97}$ | $\mathbf{97.60 \pm 2.16}$ |

Table 3: SHD averaged over 20 replications with associated 95% confidence intervals of the various models compared on the four datasets ($n = 100$). All algorithms were ran for approximately the same CPU-time and the significantly best performing models are given in **bold**. (*Lower is better.*)

number of datapoints. The issue arises as the likelihood features both in the informed proposals and in the MH step. A viable solution to this problem consists in the possibility of coupling PARNI-DAG with optimal sub-sampling MCMC procedures, such as the ones presented in Korattikara et al. [2014] and Maclaurin and Adams [2014] (FireflyMC). These limitations associated with the PARNI-DAG proposal provide us with interesting future research directions, aimed at improving its scalability and capability to handle more complex data.

## 6  Discussion

In this work, we proposed a novel MCMC sampler, PARNI-DAG, for a fully Bayesian approach to the problem of structure learning. PARNI-DAG samples directly from the space of DAGs and introduces improvements in terms of computational complexity and MCMC mixing, stemming from the nature of its adaptive random neighborhood informed proposal. This proposal facilitates moves to higher probability regions, and is particularly useful when dealing with high-dimensional settings with a high number of nodes, as demonstrated in the experimental section. In future work, we will consider adding global moves to the PARNI-DAG proposal, which re-generates a new DAG model every few iterations, to specifically tackle settings with highly correlated structures.

In addition, we are also interested in studying PARNI's theoretical mixing time bounds. In fact, although the mixing time bound for random walk proposals has been largely studied, similar results for the class of locally informed proposals are relatively under-developed, due to the complexities arising in the proposal distributions. Most results on mixing time in discrete sample spaces focus on the problem of Bayesian variable selection. It has been shown by Yang et al. [2016] that a random walk proposal (specifically the add-delete-swap proposal) can achieve polynomial-time mixing under mild conditions on the posterior distribution. Under similar conditions, Zhou et al. [2022] has shown that the mixing time of the Locally Informed and Thresholded proposal (LIT) does not depend on the number of covariates. The mixing time bound of MCMC samplers in Bayesian structure learning settings is notoriously harder to study, due to the higher complexity of the sample space of DAGs. The recent work of Zhou and Chang [2023] has shed light on the analysis of mixing time bounds for Bayesian structure learning as a generalisation of the results in Yang et al. [2016]. Zhou and Chang [2023] has proven that the mixing time of the Random Walk Greedy Equivalent Search (RW-GES) proposal is at most linear in the number of covariates and the number of datapoints. Moreover, they also presented the necessary conditions for posterior consistency in Bayesian structure learning. It has not been formally proven yet that the informed proposal achieves faster theoretical mixing time compared to the random walk proposal in the context of Bayesian structure learning, although the empirical experiments we conducted suggest so. Similarly, it is highly likely then that the PARNI-DAG proposal can also achieve dimension-free mixing, such as the LIT proposal. We leave the topic of theoretical mixing time bounds of the PARNI proposal in different applications (e.g., Bayesian variable selection, Bayesian structure learning) to future research.

## Acknowledgements

SL is supported by EPSRC research grant EP/V055380/1. AC's research is funded by the Defence Science and Technology Laboratory (Dstl) which is an executive agency of the UK Ministry of Defence providing world class expertise and delivering cutting-edge science and technology for the benefit of the nation and allies. The research supports the Autonomous Resilient Cyber Defence (ARCD) project within the Dstl Cyber Defence Enhancement programme.

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
