# OpenReview forum: "Structure Learning with Adaptive Random Neighborhood Informed MCMC"
_NeurIPS.cc/2023/Conference — NeurIPS 2023 poster_

### Official Review · Reviewer_PZBP · 2023-07-01

**Soundness:** 4 excellent
**Presentation:** 4 excellent
**Contribution:** 3 good
**Rating:** 7
**Confidence:** 4

**Summary:**

By adding various elaborations to the state-of-the-art Markov chain Monte Carlo inference algorithm, this paper achieves an efficient and effective Bayesian network inference algorithm. Bayesian networks are one of the main tools of machine learning with a long history. In general, their learning is known to be a computationally hard problem, but efficient inference algorithms based on Markov chain Monte Carlo have been actively studied. Section 2 describes the problem setup of a linear functional Bayesian network with a normally distributed weight matrix as a concrete example. Section 3.1 introduces the baseline algorithm as a state-of-the-art inference algorithm for Bayesian networks, applying the method of Liang et al. [2022]  for Bayesian variable selection problems to Bayesian networks. In Sections 3.2 through 3.4, the authors' further elaborations are carefully described step by step. Section 3.2 shows how to effectively adjust the neighborhood of the proposals, inspired by Liang et al. [2022]  for Bayesian variable selection problems. Section 3.3 introduces an effective device for (nested) sequential sampling of neighborhoods to eliminate problems that can arise in the DAG estimation problem, but not in Bayesian variable selection. Section 3.4 presents a method for properly restricting the neighborhoods, which tend to be enormous.

**Strengths:**

- This paper is a solid proposal for possible reasonable improvements in the Bayesian network inference problem, with broad coverage of the latest developments in the surrounding fields of Bayesian machine learning.
- The text is very detailed so that a wide variety of readers (from beginners to experts) can follow the history and the latest developments in the field.
- The code is provided in such a way that the proposed algorithm can be easily followed up by subsequent research, which is a very significant contribution to the field.

**Weaknesses:**

The candidate weaknesses listed below are based on questions I had during my initial peer review. As my misconceptions are resolved, they may cease to be weaknesses.

- The paper is somewhat unclear in its claims about the mixing time analysis (or empirical observation) of the proposed MCMC algorithm, although there are several mentions of it (e.g., Line 13, 288, 365).

**Questions:**

Thank you for sharing this very interesting paper. This paper has a lot of devices to help the reader understand. I really enjoyed reading about many of the parts. On the other hands, for some parts, I also worry that perhaps I am underestimating the value of this paper due to my own lack of understanding. Therefore, I would like to present some arguments below to improve my own understanding. Authors do not need to respond to the inconsequential ones, but if there is anything that authors infer that my understanding is lacking, it would be very helpful if you could respond.

- I find the author's argument against mixing times in the proposed algorithm somewhat unclear.

In this paper, there are several mentions of mixing (e.g., Line 13, 288, 365). As the author states "PARNI-DAG quickly converges to high-probability regions (Line 11),” I interpret these claims as "the proposed algorithm achieves more rapid mixing times or shorter arrival time expectations to regions of high probability (note: these two claims are equivalent (Theorem 1.4 of [*]))”.

[*] Yuval Peres and Perla Sousi. Mixing times are hitting times of large sets. Journal of Theoretical Probability, 28(2):488–519, 2015.

However, I think that in general (especially when it depends on input data) analyzing or empirically observing MCMC mixing times is not an easy task.
To my knowledge, there are not many examples of MCMC mixing times being analyzed for many problems in general. The exceptions are the following few problems:
- Bayesian variable selection:  Yun Yang, Martin J. Wainwright, and Michael I. Jordan. On the computational complexity of high-
dimensional Bayesian variable selection. The Annals of Statistics, 44(6):2497 – 2532, 2016.
- Bayesian community detection: Bumeng Zhuo and Chao Gao. Mixing time of Metropolis-Hastings for Bayesian community detection. Journal of Machine Learning Research, 22:10:1–10:89, 2021,
- Structure learning with directed acyclic graph: [+] Quan Zhou and Hyunwoong Chang. Complexity analysis of Bayesian learning of high-dimensional  DAG models and their equivalence classes, Annals of Statistics, 2023 (or arXiv:2101.04084).

Fortunately, the DAG inference problems addressed in this paper seem to be able to guarantee polynomial-time mixing times for certain algorithms (Theorem 6 of the above literature [+]).
In this light, what observations can be made about the mixing time of the proposed MCMC algorithm?
Can existing results for mixing times be easily applied to the proposed algorithm? Or is theoretical analysis of the proposed algorithm for mixing time a future issue? Also, as we can observe from the experimental results, can the proposed algorithm speed up the mixing time by an order of magnitude that depends on the problem size? Or is it an improvement of constant orders?
Fortunately, theoretical analysis of MCMC mixing time for DAGs seems to be in progress, so we hope that the author's mention of these issues (separating what is known from what is not known) will provide important insights for the reader.

**Limitations:**

As discussed in the above Questions, I am not sure the (theoretical) guarantee of mixing time of the proposed MCMC algorithm.

---

> ### Author Rebuttal · Authors · 2023-08-08
>
> We are thankful for the reviewer’s positive comments about the clarity of exposition and the developments brought about by our work. We proceed by answering the reviewer’s clarifying questions about mixing time here below.
>
> In some parts of the paper, we use the wording ‘mixing time’ to describe the empirical mixing performance observed in the output of simulated Markov chains, rather than the formal theoretical mixing time property of the MCMC algorithm. These numerical results (Section 4) suggest that the PARNI-DAG proposal is very likely to have a faster theoretical mixing time compared to the add-delete-reverse proposal, even though this has not been formally proved. Investigating the mixing time bounds of the PARNI(-DAG) proposal is a very interesting topic, but currently beyond the scope of this work. Nonetheless, to acknowledge this, we have included a brief discussion in the new version of the paper.
>
> Although the theoretical mixing time bound for random walk proposals has been largely studied, similar results for the class of locally informed proposals (including PARNI and PARNI-DAG) are relatively under-developed, due to the complexities arising in the proposal distributions. Most results on mixing time in discrete sample spaces focus on the problem of Bayesian variable selection. It has been shown in [1] that a random walk proposal (specifically the add-delete-swap proposal) can achieve polynomial-time mixing under mild conditions on the posterior distribution. Another theoretical result in [2] shows that the informed proposals can achieve dimension-free mixing time bounds under the same mild conditions. Under the conditions that posterior mass concentrates on a small set of models and the chain starts at a model close enough to the underlying “true” model, it has been shown in [2] that the mixing time of the Locally Informed and Thresholded proposal (LIT) does not depend on the number of covariates.
>
> Notoriously, the mixing time of MCMC samplers in Bayesian structure learning settings is harder to study, due to the higher complexity of the DAGs sample space. Recent work [3] has shed light on the analysis of mixing time bounds for Bayesian structure learning as a generalisation of the result from [1]. In [3], the authors showed that the mixing time of the Random Walk Greedy Equivalent Search (RW-GES) proposal is at most linear in the number of covariates and the number of datapoints. Moreover, they also presented the necessary conditions for posterior consistency in Bayesian structure learning. Proving that the informed proposal can achieve faster mixing time than the random walk proposal (as [2] did for the case of Bayesian variable selection) in Bayesian structure learning settings is still an ongoing area of research.
>
> We conclude by mentioning that, as part of a separate ongoing research, we are currently studying the mixing time of the PARNI proposal in comparison to other state-of-the-art schemes on various applications. From the empirical results in [4], the PARNI proposal has faster empirical mixing time compared to the LIT proposal, and it is highly likely that the PARNI proposal can also achieve dimension-free mixing like the LIT proposal on Bayesian variable selection. Considering the theoretical results in [2], it appears feasible to generalise them to the Bayesian structure learning setting, but we leave this for future research.
> We are happy to include in the new version of the paper a discussion about the theoretical mixing time of the PARNI-DAG proposal and how it relates to the work mentioned above. The discussion also mentions possible methods (e.g., spectral gap, canonical path analysis and drift-and-minorization methods) that can be employed to find the theoretical mixing time bound of PARNI-DAG on structure learning problems.
>
> We look forward to hearing back from you, \
> The Authors
>
>
> [1] Yun Yang, Martin J. Wainwright, and Michael I. Jordan. On the computational complexity of high- dimensional Bayesian variable selection. The Annals of Statistics, 44(6):2497 – 2532, 2016.
>
> [2] Zhou, Q., Yang, J., Vats, D., Roberts, G.O. and Rosenthal, J.S., 2022. Dimension-free mixing for high-dimensional Bayesian variable selection. Journal of the Royal Statistical Society Series B: Statistical Methodology, 84(5), pp.1751-1784.
>
> [3] Quan Zhou and Hyunwoong Chang. Complexity analysis of Bayesian learning of high-dimensional DAG models and their equivalence classes, Annals of Statistics, 2023 (orarXiv:2101.04084).
>
> [4] Liang, X., Livingstone, S. and Griffin, J., 2022. Adaptive random neighbourhood informed Markov chain Monte Carlo for high-dimensional Bayesian variable Selection. Statistics and Computing, 32(5), p.84.

---

> > ### Comment · Reviewer_PZBP · 2023-08-17
> > **I appreciate the author's very detailed and helpful responses.**
> >
> > I appreciate the author's very detailed and helpful responses.
> >
> > The detailed additional explanation for the MCMC mixing time exceeded my expectations. All my concerns have been addressed. Thank you very much. I am sure that these explanations will satisfy the more expert readers, although I believe that the original paper also entertained a diverse audience ranging from newcomers to the field to experts in Bayesian networks with a long history of research.
> >
> > I would keep the score unchanged from my initial positive impression. Thank you for sharing a solid and important paper.

---

> > > ### Author Response · Authors · 2023-08-17
> > > **Response to Reviewer PZBP**
> > >
> > > Dear Reviewer PZBP,
> > >
> > > We are very happy the response has clarified any outstanding concerns about the work. We are once again thankful for the positive comments, but mostly for the interesting discussion these have sparked about mixing time bounds of PARNI-like proposals. This will turn out to be very useful also for related ongoing (and future) research on the topic.
> > >
> > > We remain available in case any additional query arises!
> > >
> > > Kindest Regards, \
> > > The Authors

---

### Official Review · Reviewer_1zPb · 2023-07-06

**Soundness:** 3 good
**Presentation:** 2 fair
**Contribution:** 1 poor
**Rating:** 4
**Confidence:** 2

**Summary:**

This paper proposes PARNI-DAG, a new MCMC method for sampling Directed Acyclic Graphs (DAGs) that can be used for the problem of structure learning under observational data. PARNI-DAG builds on top of PARNI, and similarly uses locally informed, adaptive random neighborhood with an efficient point-wise implementation, but introduces additional improvements include: pre-tune sampler parameters using a skeleton graph derived from other methods, augmenting the search neighborhood with an edge-reversal move, and neighborhood thinning to improve computational efficiency. Experiments on some toy datasets demonstrate advantage over existing baselines.

**Strengths:**

The proposed method seems to be an effective adaptation of PARNI to do Bayesian learning on DAGs, and outperfoms a few (classical) baseline methods on a few toy benchmarks.

**Weaknesses:**

I am not an expert in this field, but it seems to me this paper is largely adapting the existing PARNI method for Bayesian variable selection for DAG learning, and combining various techniques that are already present in the current literature on top of PARNI. It would be helpful if the authors can make a more compelling case for the contributions of the paper over existing work.

For example, in L62-L72 contribution, it seems up until L69 it is just describing what PARNI already has. The procedure for pre-tune sampler parameters and do warm start also seem like a trivial application of exsiting method. In Section 3 titled "The novel PARNI-DAG" proposal, the entire Section 3.1 seems to be just a recap of wht PARNI already has. Section 3.2 is mostly a recap of Kuipers et al. [2022]. Section 3.3 introduces the reversal neighborhood but this is also done in for example partition MCMC. Section 3.4 also seems like a simple adaptation of the thinning procedure that is already present in PARNI. In fact a large part of Section 3 seems to belong to background review rather than a description of a novel method. It would be helpful if the authors can reorganze this way and also clearly state what exactly is the novel contribution of PARNI-DAG.

**Questions:**

See weaknesses

**Limitations:**

The authors did not discuss limitations

---

> ### Author Rebuttal · Authors · 2023-08-08
>
> We thank the reviewer for the time spent reviewing our submission. In what follows, we address concerns about novelty and originality of the work.
>
> The modifications made to the original PARNI proposal, in order to adjust the sampler from a pure variable selection setting to the more complex structure learning one, are several, and non-trivial. Applying a vanilla PARNI proposal to a DAG learning problem results in sub-optimal performance and particularly poses serious concerns about the sampler’s scalability, due to the complexity of DAGs space. For this reason, the adjustments we introduce are necessary to make the PARNI sampler available to the structure learning community to use.
>
> We proceed by addressing some of the specific points raised by the reviewer:
>
> •	In Section 3.2, the procedure we introduce is fundamentally different from the one described in Kuipers, J. et al. (2022). While they make use of the PC algorithm to restrict the starting search space, we instead utilise it to compute warm-start estimates (instead of the computationally intractable Rao-Blackwellised estimates used in original PARNI proposal) for the posterior edge probabilities $\pi (\gamma_{ij} = 1)$. While the former procedure introduces bias due to true positive edges deletion, ours does not, as the sampler can potentially still revert back and target the true posterior distribution.
>
> •	In Section 3.3, we describe the reversal neighbourhood construction, introduced to improve the sampler’s mixing. Although the idea of a reversal neighbourhood features also in the classical structure MCMC sampler, designing the reversal neighbourhood for the PARNI-DAG proposal is a relevant modification - the way the reversal neighbourhood is constructed in a vanilla PARNI proposal is trivial and would result in extremely slow mixing.
>
> •	Section 3.4 introduces the parameter adaptation scheme that controls the neighbourhood size and can thus significantly reduce the computational cost. The novel PARNI-DAG’s adaptation scheme comes with a completely different objective from the classic PARNI proposal, as it attempts to control the neighbourhood size directly.
>
> Without these adjustments, vanilla PARNI proposal would not be efficaciously applicable to DAG learning problems, as it would suffer from extremely slow mixing issues.
>
> We look forward to hearing back from you,\
> The Authors

---

> > ### Comment · Reviewer_1zPb · 2023-08-19
> > **Thanks for the response**
> >
> > I thank the authors for the response. However after reading the response my confusion still remains. For example the authors mentioned that `Applying a vanilla PARNI proposal to a DAG learning problem results in sub-optimal performance and particularly poses serious concerns about the sampler’s scalability` but it's not immediately clear to me what they mean by `sub-optimal performance` and what are the `serious concerns`. And with all the additional explanations this work still seems like a simple adaptation of existing techniques.
> >
> > But as I mentioned in the initial review, I am not an expert in this area. I will keep my score as is since the rebuttal does not seem convincing to me (and in may places seems to be making claims without any explanations/support). But it is possible that I do not fully grasp the challenges the authors have to overcome in the relevant adaptions, and I would be happy to hear feedback from other reviews.

---

> > > ### Author Response · Authors · 2023-08-21
> > > **Response to Reviewer 1zPb**
> > >
> > > Dear Reviewer 1zPb,
> > >
> > > Thank you for your reply. We would just like to clarify the specific points raised in the response that might be source of confusion.
> > >
> > > - By ``serious concerns`` in relation to vanilla PARNI sampler’s scalability, we mean both its **higher computational costs** and its **slower mixing**. Issues concerning the higher computational costs of vanilla PARNI are addressed via the changes introduced in Section 3.2 (warm-up PEPs estimates) and 3.4 (directly controlling neighbourhood size). As an example relative to the changes implemented in section 3.2 alone, vanilla PARNI, where the PEPs are computed adaptively via Rao-Blackwellised estimates, scales linearly in the number of edges and quadratically in the number of nodes. PARNI-DAG instead scales strictly less than that, with computational savings’ attributable to the use of warm-up PEPs estimates that depend on the starting search space obtained from the PC algorithm. Section 3.3 instead deals with slow mixing issues.
> > >
> > > - By ``sub-optimal performance`` we mean that vanilla PARNI yields a significantly lower accuracy (in addition to longer computational time) in DAG learning tasks compared to PARNI-DAG (and closer to ADR’s one). This is naturally attributable to its slower mixing properties mentioned in the point above.
> > >
> > > Kindest Regards, \
> > > The Authors

---

### Official Review · Reviewer_xD8f · 2023-07-07

**Soundness:** 3 good
**Presentation:** 3 good
**Contribution:** 2 fair
**Rating:** 5
**Confidence:** 4

**Summary:**

This paper presents a Markov chain Monte Carlo (MCMC) sampler adapted from previous PARNI sampler, called PARNI-DAG, which is designed for Bayesian structure learning under observational data. The authors assume causal sufficiency and target the posterior distribution on Directed Acyclic Graphs (DAGs). The proposed PARNI-DAG mainly relies on the PARNI sampler and modify it to suite the purpose of structure learning, including 1. warm-start of neighbourhood sampling probability 2. reversal neighbourhood step and 3. adaptive neighbourhood skipping probability.

The main contributions of the paper are

1. A warm-start procedure of the sampler's parameters that exploits skeleton graphs derived through constraint-based or scoring-based algorithms, ensuring better scalability and mixing property with the number of nodes.

2. A reverse step to avoid getting trapped in the local mode.

3. An adaptive skipping probability such that not all intermediate neighbourhood sampling steps are executed.

Empirically, the author demonstrates the advantage of PARNI-DAG using real-world examples, which shows advantages compared to the previous MCMC methods.

**Strengths:**

## Originality
The originality of the paper lies in the development of the PARNI-DAG algorithm, which combines the Point-wise Adaptive Random Neighborhood Informed (PARNI) proposal with new features specifically designed for structure learning in the space of Directed Acyclic Graphs (DAGs). The proposed algorithm addresses the challenges of mixing and convergence in high-dimensional settings, which is not adequately addressed by existing MCMC methods for structure learning. This work does not provides a completely new sampling algorithm, instead, modify the existing approaches to suite the purpose of structure learning.

## Clarity
The paper is logically structured with clear presentation of the modifications made to the PARNI proposal. The authors have made efforts to provide intuitive explanations and motivate their choices in the development of the PARNI-DAG algorithm. The paper is easy to follow, and the appendices provide additional details on the derivations and calculations.

## Significance
The significance of the paper lies in its potential impact on the field of structure learning and causal discovery. The PARNI-DAG algorithm aims to address the challenges of mixing and convergence in high-dimensional settings. The algorithm's improved performance over existing MCMC methods for structure learning should make it a reasonable contribution to the field but there are some limitations, which I will elaborate in the following.

**Weaknesses:**

## Empirical experiments:
While the experimental results demonstrate the advantages of PARNI-DAG over existing MCMC methods, the experiments primarily focus on the comparison with MCMC based approach. However, in the literature review, the author also mentioned several structure learning approach. Although the main claim of this paper is the improvement over existing MCMC, it is still beneficial to include a comparison to state-of-the-art Bayesian structure learning approach like [1,2,3].

[1] Cundy, C., Grover, A., & Ermon, S. (2021). Bcd nets: Scalable variational approaches for bayesian causal discovery. Advances in Neural Information Processing Systems, 34, 7095-7110.
[2] Lorch, L., Rothfuss, J., Schölkopf, B., & Krause, A. (2021). Dibs: Differentiable bayesian structure learning. Advances in Neural Information Processing Systems, 34, 24111-24123.
[3] Geffner, T., Antoran, J., Foster, A., Gong, W., Ma, C., Kiciman, E., ... & Zhang, C. (2022). Deep end-to-end causal inference. arXiv preprint arXiv:2202.02195.

## Limitation with linear model
The proposed PARNI-DAG mainly targets at the linear model with Gaussian assumptions because the marginal probability are needed. For general nonlinear model, such integration is not tractable, which rendering the following PARNI-DAG proposal invalid. However, nonlinearity is everywhere in the real-world applications. To make the paper stronger, the author should discuss the implication of using linear model or demonstrates that linearity assumption does not harm the performances too much. This can be achieved by comparing some of the previous mentioned baselines [1,2,3]

## Computational complexity
Although the proposed PARNI-DAG method uses many modifications to reduce the computation cost, this approach relies on the MH step to correct the bias. It is known that MH step scales linearly with the number of datapoints, which can be a huge computational bottleneck for large dataset. I suggests the author should explicitly discuss the computational complexity or limitations, and potential approach to remove this constraints.

**Questions:**

All the questions are mentioned in the weakness section.

**Limitations:**

The author does not explicitly write a limitation section, I have made suggestions to include one like computational challenges and limitations of using linear model.

---

> ### Author Rebuttal · Authors · 2023-08-08
>
> We thank the reviewer for the time spent carefully reviewing the paper and their positive comments about its originality and clarity. We proceed by addressing the points raised in the weaknesses section.
>
> 1)	Empirical Experiments
>
> We have implemented DiBS and DiBS+ (Lorch, L. et al., 2021), and DECI (Geffner, T. et al., 2022) models on the experimental setups of Section 4.2. However, we have noticed that, perhaps not so surprisingly, these variational methods struggle to detect edges in these low-sample regimes, as they require a stronger signal (or larger samples, as featured in their experiments) to perform well. In the pdf file attached to the global response, we have added a table reporting these methods’ average SHD and number of edges detected in the first three datasets (ecoli, magic-niab, magic-irri), as in the last dataset (arth150), characterised by large number of nodes, they are significantly slow and run into numerical instability issues.
>
> 2)	Limitations of Linear Model
>
> We have added a new ‘limitations’ section in the manuscript, where we discuss both implications of the linearity assumption and computational complexity of PARNI-DAG. Linearity is naturally restrictive, particularly when the objective of the inference is the joint posterior $p(G, \theta | D)$ rather than the marginal posterior $p(G | D)$ that we focus on, where model misspecification is a threat only when it hampers edges identifiability - meaning that some edges could potentially still be detected in presence of non-linear relationships. Unfortunately, the PARNI-DAG proposal is not directly extendable to Additive Noise Models (ANM) (Hoyer, P. et al., 2008) where the closed-form marginal likelihood does not exist. As part of a separate, but related, research project we have developed a new PARNI proposal coupled with (approximate) Laplace approximation (Rossell, D. et al, 2021) suitable in contexts where the marginal likelihood is not closed form. We have shown that it works well empirically in Bayesian variable selection tasks in generalised linear models and survival models. This provides a solid base to potentially extend the PARNI-DAG sampler to ANM with approximate marginal likelihood, although we leave this for future research.
>
> 3)	Computational Complexity
>
> The other limitation of the PARNI-DAG proposal pointed out by the reviewer, and discussed in the newly added limitation section, is the computational cost associated with the likelihood evaluation, which scales at least linearly in the number of datapoints. The issue arises as the likelihood features both in the informed proposals and in the MH step. A potential solution to this problem, that we discuss in the new section, regards the possibility of coupling PARNI-DAG with sub-sampling MCMC procedures, such as the ones presented in Korattikara, A. et al. (2014) and Maclaurin, D. & Adams, R. (2015) (FireflyMC). We are actively investigating this topic as a part of a separate ongoing research project.
>
> We look forward to hearing back from you, \
> The Authors
>
>
> [1] Hoyer, P., Janzing, D., Mooij, J.M., Peters, J. and Schölkopf, B., 2008. Nonlinear causal discovery with additive noise models. Advances in neural information processing systems, 21.
>
> [2] Rossell, D., Abril, O. and Bhattacharya, A., 2021. Approximate Laplace approximations for scalable model selection. Journal of the Royal Statistical Society Series B: Statistical Methodology, 83(4), pp.853-879.
>
> [3] Korattikara, A., Chen, Y. and Welling, M., 2014, January. Austerity in MCMC land: Cutting the Metropolis-Hastings budget. In International conference on machine learning (pp. 181-189). PMLR.
>
> [4] Mclaurin, D. and Adams, R., P., 2015, Firefly Monte Carlo: Exact MCMC with Subsets of Data. Proceedings of the 24th International Conference on Artificial Intelligence (pp. 4289–4295). IJCAI'15.

---

> ### Comment · Reviewer_xD8f · 2023-08-16
>
> Thanks for the authors' feedback regarding the additional experiments. They addressed my concerns and promised to add new limitation section. So I will keep my current evaluation.

---

> > ### Author Response · Authors · 2023-08-17
> > **Response to Reviewer xD8f**
> >
> > Dear Reviewer xD8f,
> >
> > Thank you for your reply. We remain available in case any further clarification is needed.
> >
> > Kindest Regards, \
> > The Authors

---

### Author Rebuttal · Authors · 2023-08-08

We are thankful to the reviewers for the time spent going through our submission and for their insightful comments, as we believe these have significantly contributed to improving the paper.

In particular, we have addressed Reviewer xD8f’s request for additional experimental comparison by implementing the variational methods DiBS, DiBS+ and DECI (Lorch, L. et al., 2021; Geffner, T. et al., 2022) on the experimental setups of Section 4.2, and showed that these are not particularly well-suited for these types of low data regimes (a table of the results can be found in the pdf attached this global response). We have also added a new section in the paper discussing the limitations of the linearity assumption and the computational complexity of PARNI-DAG together with potential ways to alleviate this (sub-sampling MCMC solutions), which we are currently investigating.

We have addressed Reviewer 1zPb’s concerns about the originality of the work, by stressing how PARNI-DAG brings about several relevant adjustments to the vanilla PARNI proposal that makes the methods applicable to the higher complexity of DAG spaces. We have stressed also that vanilla PARNI would yield sub-optimal performance and would pose serious scalability concerns in structure learning settings.

Finally, in response to Reviewer PZBP’s questions, we have included in the new version of the paper a brief discussion on the theoretical mixing time of the PARNI and PARNI-DAG proposals. As mentioned in the last paragraph of the individual response to Reviewer PZBP, as part of a separate ongoing project, we are currently studying mixing time bounds of the classic PARNI proposal in Bayesian variable selection settings, and plan to potentially extend this study to Bayesian structure learning settings in the future, although this is notoriously a harder problem.

We hope we have adequately responded to all the concerns regarding the paper, and we look forward to engaging with the reviewers during the interactive discussion period, to respond to any outstanding queries.

Kindest Regards, \
The Authors

---

### Decision · Program_Chairs · 2023-09-21

**Decision:**

Accept (poster)

**Comment:**

While we didn't reach full consensus among the reviewers, two of the three reviewers were satisfied with the author responses and recommend acceptance.  The authors are strongly encouraged to make the changes the proposed in the replies to these reviewers.  There was good back-and-forth between the third reviewer and the authors, although the third reviewer never replied to the last author reply.  This meta-reviewer found that author reply satisfactory and so is recommending acceptance.  Authors are encouraged to take into careful consideration the comments of that reviewer as well in preparing the final version of the paper.